# Decoupling Urban and Non-Urban Landslides for Susceptibility Mapping in Transitional Landscapes, a Case Study from Transitional Landscapes in Southwestern Constantine, Algeria

Zakaria Matougui<sup>1</sup>, Yacine Mohamed Daksi<sup>1</sup>, Mehdi Dib<sup>1</sup>, Chaouki Benabbas<sup>1</sup>

<sup>1</sup>Researcher, Centre de Recherche en Aménagement du Territoire (CRAT), Campus Zouaghi Slimane, Route de Ain elBey, 25000 Constantine, Algérie

Correspondence to: Zakaria Matougui (zakaria.matougui@crat.dz)

**Abstract.** This study develops a framework for decoupling and investigating urban and non-urban landslide mechanisms, focusing on Constantine, Algeria, a city with complex topography and high landslide susceptibility. The region presents a heterogeneous landscape, where dense urban zones coexist with bare rural areas, influencing slope stability differently. A landslide inventory of 184 events was compiled and classified into urban and non-urban categories. Using geospatial data (topography, hydrology, landcover, lithology) and machine learning models (Random Forest, XGBoost, LightGBM, Multi-Layer Perceptron, and Logistic Regression), landslide susceptibility maps were generated for three datasets: urban, non-urban, and mixed. Model performance was assessed using cross-validation and evaluation metrics (ROC-AUC, F1-score, precision, recall), while SHAP analysis provided insights into factor importance. The results reveal distinct landslide drivers across environments. In urban areas, landslides are primarily influenced by aspect, slope, and proximity to streams, while distance to roads plays a lesser role, likely due to engineered slopes and drainage infrastructure. In non-urban areas, distance to roads is the most critical factor, highlighting the destabilising effects of road cuts in rural landscapes. Slope and proximity to streams remain key determinants, with lithology playing a more significant role in naturally driven failures. This study underscores the importance of context-specific landslide modelling and the potential biases of using mixed urban and non-urban inventories. The findings provide actionable insights for targeted mitigation, land-use planning, and infrastructure design. By distinguishing between urban and non-urban landslides, this research bridges critical gaps in understanding landslide dynamics across diverse landscapes.

#### Introduction

In the face of rapid urbanization and the rarefaction of land suitable for construction, particularly in hilly and mountainous regions, ensuring the safety of infrastructures has become a crucial challenge due to the prevailing soil instabilities and disorders. Accurately assessing these hazards is essential for informed spatial planning, enabling sustainable economic development and safeguarding the well-being of communities. This context, which combines expanding urban areas over hilly

regions, raises a number of challenges related to the assessment of soil instability due to the interplay of anthropogenic and natural factors. Urban development alters slope stability through construction activities, modifications to natural drainage (surface and subsurface water circulation) and increased loads, while deforestation and land use changes accelerate erosion and reduce natural stabilization (Benabbas, 2006; Hadji et al., 2013). Furthermore, the higher density of infrastructures in urban areas significantly alters and weakens the natural landscape, potentially intensifying pressure on slope stability and thereby increasing susceptibility to slope failures (Carrión-Mero et al., 2021; El Kechebour, 2015). In contrast, slope stability in rural regions is shaped predominantly by natural factors (Carrión-Mero et al., 2021), such as heavy rainfall, seismic events, and lithological evolution, leading to different instability mechanisms compared to urban settings.

Northern Algeria, with towns built on largely loose soil (marl and clay), often unstable and particularly vulnerable to landslides. The city of Constantine, as some geology and geotechnical specialists often referred to as "an open sky museum for landslides" exemplifies this problem (Bougdal et al., 2007). As Algeria's third-largest urban area and the most important in the eastern region, Constantine frequently experiences landslides that not only affect densely populated urban areas but also neighbouring rural regions (Mezerreg et al., 2019). This duality of urban and non-urban impacts makes Constantine a prime location for studying the interplay of factors driving landslides in different environments. Rapid urbanization in regions like Constantine often results in incomplete or insufficient geospatial data, complicating the integration of urban characteristics into predictive models, particularly in heterogeneous and transitional zones between urban and non-urban areas (buffer zones). This makes it challenging to accurately capture the spatial variability of landslide mechanisms. Moreover, population exposure, environmental degradation, and the increased strain on mitigation infrastructure amplify the risks, underscoring the need for targeted and context-specific hazard assessments (Achour et al., 2017; Manchar et al., 2018; Mentouri et al., 2018).

Although extensive research has been carried out on landslide hazard assessment in urban areas (Bathrellos et al., 2009; Huang et al., 2023; Pascale et al., 2010, 2013), these studies generally examine urban environments in isolation. They rarely investigate how urban and non-urban processes differ, particularly in transitional zones where the two settings interact. At the same time, the broader international literature shows that machine learning (ML) has become a widely adopted tool for landslide susceptibility mapping. However, most applications still treat the landscape as a homogeneous entity, without explicitly distinguishing between urban and non-urban contexts.

Early work illustrates this limitation. For instance, (Caniani et al., 2008) applied artificial neural networks in Potenza, Italy, but without differentiating urban from non-urban landslides either in the inventory or during modelling. More recent studies have followed a similar approach. (Islam et al., 2025) applied a hybrid ML model to a rapidly urbanizing area in Bangladesh, and (Luo et al., 2025) assessed landslide hazards in the central Guizhou urban agglomeration using SVM, DNN, and bagging algorithms; in both cases, inventories did not explicitly separate urban from non-urban events.

This limitation is critical because urban landslides are often shaped by anthropogenic factors, including slope modification, drainage alteration, construction practices, and infrastructure density, that differ markedly from the natural drivers that dominate in non-urban areas. Without explicitly accounting for these differences, susceptibility models risk oversimplifying complex processes and overlooking the distinct mechanisms of slope failure across contrasting environments.

This study aims to address these challenges by bridging a significant gap in the literature: decoupling urban and non-urban instabilities for landslide susceptibility mapping. By separately analysing the unique driving factors in urban and non-urban environments, this research provides a more nuanced understanding of landslide susceptibility, thereby enhancing the accuracy and applicability of predictive models for sustainable urban and rural development. To achieve this, the study employs a comprehensive methodology integrating geospatial data, machine learning algorithms, and advanced analytical techniques. A detailed landslide inventory was constructed trough a comprehensive field observations and remote sensing imagery, enabling the classification of landslide events into urban and non-urban categories. Causative factors, including topographical, geological, hydrological, and land-use variables, were extracted from sources such as digital elevation models, geological maps, and land cover data. These datasets were standardized and converted into machine-readable formats to ensure consistency and precision. Machine learning models, including Logistic Regression, Random Forest, LightGBM, XGBoost, and Multi-Layer Perceptron, were trained separately for urban, non-urban, and mixed datasets, with hyperparameters optimized using Bayesian techniques. Model performance was evaluated using cross-validation and metrics such as ROC-AUC, F1score and recall while SHAP analysis was employed to interpret the relative importance of each factor. This methodological framework not only produces robust landslide susceptibility maps but also elucidates the distinct mechanisms driving instability in urban and non-urban settings, thereby addressing critical gaps in the existing literature.

#### 1. Materials and Methods

## 1.1. Study area

- The territory of the city of Constantine (**Fig. 1**) is distinguished by a complex morphology, characterized by irregular hills and often deep valleys, which considerably increase its vulnerability to landslides. The old town (three thousand years old) is built on a carbonate and karstified rocky plateau of Cenomanian-Turonian age. The Rhumel river was able to dig and impose a canyon on this limestone plateau. We think that the digging of the canyon of the rock of Constantine was dictated by the post-Pliocene (Quaternary) uplift of this massif and that this uplift is still ongoing.
- The region's Mediterranean climate, marked by hot, arid summers and mild, wet winters, leads to episodic intense rainfall events that promote slope saturation and increase the likelihood of mass movements. Furthermore, the predominance of clayrich and moisture-retentive geological formations significantly amplifies soil instability, particularly under conditions of heavy precipitation and seismic activity.
  - Constantine has been the focus of numerous studies addressing landslide susceptibility and risk. Early research by (Benaissa and Bellouche, 1999) examined the geotechnical properties of landslide-prone formations, revealing the instability of terrains within the urban area. (Guemache et al., 2011) highlighted the technical difficulties of stabilizing landslides, particularly in the case of the Sidi Rached bridge. Later works, such as those by (Bourenane et al., 2015), applied statistical methods and GIS to map susceptibility, while (Manchar et al., 2018) utilized AHP and interpretation of aerial images for hazard assessment. These

foundational studies provide a robust theoretical and methodological base for the current work, which introduces an innovative approach by decoupling urban and non-urban factors in the identification and landslide susceptibility mapping.

Rapid and often unplanned urbanization has led to increased impervious surfaces, altered drainage patterns, and intensified land use changes. The expansion of residential, commercial, and infrastructural developments into hilly areas has disrupted natural slope stabilization mechanisms, increased slope loading, further weakening soil structures. Consequently, Constantine's infrastructure, including roads, bridges, and buildings, is frequently threatened by landslides, posing significant risks to public safety and economic stability (Guemache et al., 2011; Schlögl et al., 2019).

100

105

Non-urban areas surrounding Constantine are equally vulnerable, though influenced by different factors. These regions are primarily affected by natural processes such as heavy rainfall, seismic activity, and continuous weathering of geological formations (Mounia et al., 2013). Additionally, agricultural practices, land clearing, and minor construction activities in rural zones contribute to soil erosion and slope degradation, although to a lesser extent compared to urban settings. The heterogeneous landscape of Constantine, characterized by the juxtaposition of densely built urban zones and more stable rural areas, presents a unique opportunity to study the differential impacts of urbanization on landslide susceptibility.

Figure 1 Location of the study area as hill shading to represent the topography with the road network, stream network and landslide polygons mapped.

## 1.2. Landslide inventory

110

115

Urban and non-urban landslides differ significantly in their triggers and characteristics due to varying environmental, geological, and human factors. In urban areas, landslides are primarily driven by human activities such as construction, excavation, deforestation, and poorly designed drainage systems. These actions disrupt natural slope stability and reduce soil cohesion, increasing the likelihood of landslides near infrastructure. However, urban landslides can also be triggered by natural events like heavy rainfall and earthquakes, which exacerbate existing vulnerabilities caused by urbanization. In contrast, non-urban landslides are mainly caused by natural processes such as intense rainfall, erosion, and neotectonic seismic activity, with

vegetation cover playing a vital role in stabilizing the soil. We identified these rural slope failures through a combination of direct field investigation and multi-temporal remote sensing interpretation (Alharbi et al., 2014; Varnes, 1984).

The triggering factors also vary between environments: urban landslides may result from a combination of human-induced disturbances like leaks or vibrations and natural events such as heavy rains or earthquakes (Chen and Wang, 2023; Ma and Wang, 2024). In contrast, non-urban landslides are mainly influenced by natural phenomena. Understanding these differences is crucial for developing accurate landslide susceptibility maps and implementing effective mitigation strategies tailored to each setting.

Landslide characteristics

120

140

145

The lithological diversity observed in the Constantine study area strongly influences the types of instabilities likely to occur. In the western region, predominant formations consist of thick marls and marly clays, which are particularly susceptible to water saturation. This lithological setting promotes deep rotational landslides, due to the substantial thickness of these cohesive and plastic facies, which can become extensively waterlogged, particularly on moderate to steep slopes (Manchar et al., 2018). Conversely, in the northeastern region, where shallow clays overlay a hard substratum, typically composed of indurated 130 limestone or conglomerates, planar translational landslides are common. Such movements are characteristic of stratified terrains, where a clear sliding surface forms between the superficial loose formation and the underlying rigid formation. Additionally, in areas with gentler slopes, solifluction phenomena occur, involving slow movements of saturated fine materials, exhibiting limited acceleration due to low slope inclination. Lastly, the central region presents a complex scenario: watersensitive marls coexist with densely built environments, placing additional stress on slopes. This context leads to complex 135 landslides triggered by surcharge, modified hydrological flows, and progressive slope degradation. This variety of instabilities necessitates a differentiated typological approach, integrating geological facies, slope geometry, and anthropogenic factors to effectively map and anticipate slope movements (Hungr et al., 2014).

Landslide inventory classification

The landslide inventory was constructed with the objective of distinguishing between urban and non-urban slope failures while maintaining methodological consistency across the study area. Classification was carried out by overlaying mapped landslide polygons with the land-cover dataset. An event was defined as urban when its polygon intersected built-up zones or close to infrastructure. Conversely, landslides located entirely outside these zones and in areas characterized by bare or agricultural land cover were classified as non-urban. *Landslide Inventory in Urbanized Areas* 

To establish a comprehensive landslide inventory within the urbanized portions of the study area, landslide identification was primarily based on in-situ observations. A sequence of indirect indicators was employed to detect ground instabilities associated with landslides. These indicators were systematically observed across various infrastructures, including buildings, roads, and sewerage systems. Given the high density of construction in urbanized areas, the utilization of high-resolution satellite imagery for delineating landslides proved ineffective. Consequently, surveys and field observations were the primary tool adopted in this study.

To differentiate a landslide from phenomena such as uneven settlement or swelling, an event was only included in the inventory if its vicinity exhibited multiple indirect indicators in conjunction with testimonies from inhabitants and expert opinions. This multi-faceted approach ensured the reliability and accuracy of the landslide inventory. The indirect indicators utilized in the landslide inventory for urbanized areas include:

Cracks in Buildings and Roads:

Buildings: Horizontal or diagonal cracks in floors or walls were considered key indicators of potential landslide activity (Fig. 2 a-b).

**Roads:** Longitudinal and transverse cracks in road surfaces, pavements, and car parks were identified as significant indicators of slow-moving landslides (**Fig. 2-a**).

Uneven Settlement in Buildings: Uneven ground settlement associated with subsurface instability manifested as uneven floors within structures (Fig.2 b&f).

**Deformation of Roadways:** The presence deformations in roadways such us misalignments, bulges, and depressions suggest, if associated with other indirect indicators in the road vicinity, the presence of a slow landslide.

**Underground utility damage:** Damage to underground utilities, including water and sewage systems, when correlated with other nearby indirect indicators, was attributed to soil movement.

**Failing retaining infrastructures:** The failure of retaining walls and other soil reinforcement structures served as indicators of landslide activity (**Fig. 2 c-d**).

Leaning trees and other structures: Trees, electricity poles, or other structures that exhibit leaning from their original vertical positions were considered signs of slope movement or ground instability, especially when accompanied by other indirect indicators (Fig. 2 b-e).

Landslide Inventory in Non-Urbanized Areas



In non-urbanized areas, the landslide inventory was constructed primarily through remote sensing interpretation, complemented by selective field verification. Mapping was guided by morphological criteria following the principles of (Varnes, 1984) whereby typical geomorphic signatures of mass movements, such as scarps, displaced material, surface cracks, and toe bulges, were visually identified and delineated. This geomorphological approach ensured that landslides in rural sectors were consistently captured and provided a methodological counterpart to the field-based strategy applied in urban areas.

Each landslide was delineated by mapping the entire affected surface, from the main scarp to the toe, thus incorporating both the source and accumulation zones. Due to the predominance of clays and marls overlying hard limestone or conglomerate formations, most slope movements in the study area are characterized by moderate displacements and relatively small deposition areas. The inventory including urban and non-urban landslides were compiled during a comprehensive field and remote sensing interpretation survey conducted between June and December 2024.

This systematic approach to inventory creation, leveraging multiple indirect indicators alongside direct observations and expert assessments, ensures a robust and reliable dataset for subsequent landslide susceptibility mapping. By focusing on both

structural and natural signs of instability, the inventory comprehensively captures the multifaceted nature of urban landslide phenomena, effectively distinguishing them from other ground movement events (Bornaetxea et al., 2018).

To further characterize the mapped landslides, descriptive statistics were calculated for the urban, non-urban, and mixed inventories (**Table 1**). The urban dataset comprises 123 landslides totalling 18.4 ha, while the non-urban dataset includes 61 landslides covering 21.2 ha. Combined, the mixed inventory contains 184 landslides with a total area of 39.6 ha. Despite its larger number of events, the urban dataset represents the smallest total area.

#### Table 1 Descriptive statistics of mapped landslides in the study area


| Type      | Count | Total Area<br>(ha) | Mean Area (ha) | Median Area<br>(ha) | Min Area<br>(ha) | Max Area<br>(ha) |
|-----------|-------|--------------------|----------------|---------------------|------------------|------------------|
| Non-urban | 61    | 21.24              | 0.348          | 0.111               | 0.0058           | 3.97             |
| Urban     | 123   | 18.4               | 0.1496         | 0.041               | 0.0006           | 4.37             |
| Mixed     | 184   | 39.64              | 0.215          | 0.051               | 0.0006           | 4.37             |

Esri, HERE, Garmin, (c) OpenStreetMap contributors, and the GIS user community, Source: Esri, Maxar, Earthstar Geographics, and the GIS User Community

Figure 2 Field evidence of slope instability in the study area: a) Example of cracks in buildings and roads; b) Shallow landslide scarp in an urban setting, with an adjacent demolished structure; c) Failure of a concrete retaining wall; d) Sheet-pile retaining system exhibiting lateral bulging; e) Leaning tree signalling slope deformation; f) Example of inclined buildings (© Esri, © Open Street Map contributors, and the GIS user community, Source: Esri, Maxar, Earthstar Geographics, and the GIS User Community)

#### 1.3. Landslide causative factors





Selecting appropriate landslide conditioning factors is essential for accurate susceptibility modelling. In this study, factors capturing both anthropogenic and natural influences were chosen to distinguish between urban and non-urban landslides. By analysing these factors, we isolate the distinct mechanisms driving slope failures in different environments (**Fig. 3**), enhancing model precision and interpretability.

The conditioning factors employed in this study are detailed in **Table 2**. Topographical and hydrological variables, including Elevation, Slope, Aspect, Curvature, TWI (Topographical Wetness Index) and Distance to stream, were derived from the ALOS PALSAR dataset and DEM-based calculations. These factors are crucial for characterising the terrain, as elevation and slope affect gravitational forces and mass movement potential, while aspect and curvature influence microclimatic conditions and water flow dynamics. TWI quantifies soil moisture accumulation, and distance to streams provides an indication of potential water-induced processes.

The Land Use and Land Cover information was sourced from the ESA WorldCover dataset at a 10-metre resolution. This dataset differentiates between natural vegetation and anthropogenic land uses, distinguishing between urban and non-urban areas. Additionally, the Distance to roads, obtained from OpenStreetMap, captures the influence of infrastructure on slope stability by identifying modifications in natural drainage and terrain disturbance.

Geological factors have been incorporated via a Lithology map based on expert field surveys, which provides vector data on soil and rock formations. This allows for the identification of areas with potentially fragile substrates that are more susceptible to landslides. Finally, the NDVI, calculated using Harmonised Sentinel-2 MSI data, offers a quantitative measure of vegetation cover at a 10-metre resolution. This factor is important in assessing the role of vegetation in reinforcing soil and mitigating erosion, thereby influencing overall slope stability.

Table 2 Summary of landslide conditioning factors used in this study

| Factor Type            | Source                                                    |        |
|------------------------|-----------------------------------------------------------|--------|
| Elevation (m)<br>(DEM) | ALOS PALSAR dataset (2006–2011; DOI:10.5067/Z97HFCNKR6VA) |        |
| Slope (°)              |                                                           |        |
| Aspect                 |                                                           | 30     |
| Curvature              | Digital Elevation Model (DEM) derived calculation         | metres |
| TWI                    |                                                           |        |
| Distance to stream (m) |                                                           |        |

| Land Use and Land Cover | ESA WorldCover dataset (DOI:10.5281/zenodo.5571936)                          | 10 metres |
|-------------------------|------------------------------------------------------------------------------|-----------|
| Distance to roads       | OpenStreetMap (2019)                                                         |           |
| (m)                     | opensuced viap (2017)                                                        | data      |
| Lithology               | Digital lithological map (expert field surveys)                              |           |
| Littlology              |                                                                              |           |
| NDVI                    | Harmonised Sentinel-2 MSI (median of time series between 28 March 2017 and 2 | 10 metres |
| 1,5,1                   | February 2025) (Claverie et al., 2018)                                       |           |

Figure 3 Spatial distribution of conditioning variables used in landslide susceptibility modeling for the study area including: Elevation (m); Slope (°); Distance to streams (m); Aspect (°); Curvature; TWI; Distance to roads (m); Lithology; (i) NDVI; and Land use

## 225 Density distributions of landslide conditioning factors




To understand the influence of environmental and anthropogenic factors on slope stability, probability density functions were estimated for each conditioning variable using a kernel density estimator (KDE) (Chen, 2017):

$$\hat{f}_h(x) = \frac{1}{nh} \sum_{i=1}^n K\left(\frac{x - x_i}{h}\right) \tag{1}$$

where  $\hat{f}_h(x)$ : estimated probability density at x, n the number of observations, h the bandwidth (smoothing parameter), K the kernel function (e.g., Gaussian), and  $x_i$  the individual observations.

This approach provides a smoothed representation of the distributions of landslide and non-landslide cells, while allowing urban and non-urban landslide occurrences to be analysed separately. By overlaying these density plots (Fig. 4), several insights can be obtained:

- Identifying Factor Importance: Large separations in the distributions (e.g., steep slope angles) suggest that certain factors are particularly significant in driving landslides.
- Distinguishing Urban vs. Non-Urban Patterns: Overlaying urban and non-urban curves highlights how features
  interact with landslide occurrence.
- Revealing Potential Thresholds: Certain factor ranges (e.g., slope >5° or elevation between 500 and 600 m) align with higher or lower landslide frequencies, informing both susceptibility modelling and mitigation strategies.
- Land Use: Urban landslide densities (red) concentrate heavily in built-up areas. In non-urban settings, landslides are more common in naturally vegetated or agricultural zones, indicating the influence of mostly natural triggers such as intense rainfall, erosion, and weathering. Nonetheless, agricultural practices (eg terracing, irrigation) can also modify slope geometry, soil composition, and water infiltration, which may heighten landslide susceptibility in some areas. Therefore, while urbanisation is often associated with more pronounced slope disturbance and drainage alteration, non-urban landslides remain vulnerable to both natural processes and lower-intensity human activities, highlighting that land use exerts a significant, but context-dependent, influence on slope stability.

*Lithology*: Clayey marl and marly clay stand out as common substrates for both urban and non-urban landslides. Urban areas, however, are characterized by a pronounced presence of these lithological formations, indicating that when inherently weak materials coincide with excavations, leaking infrastructure, or other human-induced alterations, the susceptibility to failure increases dramatically. Geological weaknesses such as low shear strength therefore play an important role, but are intensified in urban contexts by anthropogenic stressors.

**Aspect**: The data indicate that urban landslides tend to be more frequent on slopes facing northeast to northwest, whereas non-urban landslides show a slight preference for north-northwest orientations. Aspect often influences microclimatic conditions such as sunlight exposure and moisture retention; however, in urban settings, these natural patterns can be overshadowed by human-induced modifications to drainage, loading, and vegetation cover.






*Curvature*: Both urban and non-urban landslides appear clustered around low or slightly negative curvature values, corresponding to near-planar or mildly convex slopes. Extremely convex or concave slopes may be less frequently occupied or stabilised by vegetation, which could explain their lower landslide density. Mild curvature zones can conceal latent instabilities, especially when subjected to additional loading or inefficient drainage systems.

*Elevation*: Both urban and non-urban landslides tend to concentrate at mid-range elevations, although urban landslides may favour lower hills where city expansion is more common. Elevation influences climate (rainfall patterns, temperature) and vegetation cover, which in turn affect slope stability. However, urban development decisions, such as building on certain elevation tiers, can interact with these natural processes to increase landslide susceptibility.

Slope: In non-urban regions, landslides predominantly occur on steeper slopes ( $\geq 8^{\circ}$ ), where gravity-driven failures are more frequent in the absence of anthropogenic interventions. Urban landslides also manifest on moderate to steep slopes, but can arise on low slopes ( $

Figure 4 Density distributions of landslide conditioning factors

## 2. Methodology


This study adopts a multi-stage framework designed to disentangle the distinct mechanisms governing urban and non-urban landslides within the same broader landscape. Additionally, it aims to evaluate the potential bias introduced when relying on a purely urban or purely non-urban landslide inventory in heterogenous landscape contest. By analysing model performance

across different dataset configurations, the study assesses whether predictive capabilities differ significantly depending on the spatial context of the training data, highlighting the implications of dataset composition for susceptibility modelling. To achieve this, the methodology began with the compilation of a comprehensive mixed landslide inventory (**Fig. 5**), incorporating all documented events. This inventory was then subdivided into Urban and Non-Urban datasets based on the land use to enable more focused analyses. The *Urban* subset comprises 123 landslide events (16.5% of the total affected surface), while the *non-urban* subset contains 61 events (83.5% of the total landslides area). By segmenting the data, we can evaluate the occurrence of landslides in each environment.






Environmental factors were integrated according to the specific needs of each dataset. The *Mixed* dataset retained all covariates, including land use, to reflect the heterogeneity of the entire study area. In contrast, land use was excluded from both the Urban and Non-Urban datasets to avoid redundancy and biasing the models; these subsets inherently represent different land cover categories. This selective factor inclusion ensures that each model targets only those predictors most relevant to its respective environment.

All datasets were standardised to a 10-metre spatial resolution for consistency, converting raster layers into dataframes so that each grid cell corresponds to a single data point with associated environmental variables. Negative sample selection followed a twofold procedure: first, we applied a knowledge-based method in which experienced local geologists delineated areas that, based on comprehensive field investigations, low slopes, 50m buffer from landslides and historical records, were considered stable (**Fig. 6**); and second, within these expert-identified stable zones, we employed random sampling to generate a set of non-landslide points for model training.

Each of the three datasets (Mixed, Urban, and Non-Urban) was then subjected to a consistent analytical pipeline. Label encoding and feature scaling were applied to harmonise data for machine learning models. A multicollinearity analysis was performed to identify and remove highly correlated variables. We employed several machine learning algorithms—including LightGBM, XGBoost, Random Forest, Multi-Layer Perceptron (MLP), and Logistic Regression—with hyperparameter tuning via Bayesian Optimisation (Sun et al., 2024; Yang et al., 2023). **Table 3** summarises the key machine learning algorithms employed for landslide susceptibility mapping in this study. The table highlights each algorithm's defining characteristics alongside its suitability.

To evaluate the predictive capabilities of the models, a 10-fold cross-validation strategy was adopted. The performance metrics including accuracy, recall, F1 score and ROC-AUC (**Table 4**)—are then averaged over the ten folds and the standard deviation is calculated for a more reliable estimate of out-of-sample performance (Mas et al., 2013; Saha et al., 2020; Tang et al., 2020). Following the initial performance assessment, SHAP (SHapley Additive exPlanations) analysis was conducted to interpret the importance of individual factors in each model (Liu et al., 2024; Lundberg et al., 2018). SHAP values quantify how much each predictor contributes to moving a model's output from a baseline prediction, offering transparent insights into why certain instances were classified as landslides (or non-landslides). Finally, calibration plots were generated to compare the predicted probability distributions across different datasets. These plots gauge how well the predicted probabilities align with the actual frequencies of landslide occurrence, highlighting any systematic over- or under-confidence in the models' predictions (Gerds

et al., 2014; Lv et al., 2024). Once the predictive models were finalised, their outputs were transformed into spatially explicit susceptibility maps. Each pixel (or grid cell) in the study area was assigned a probability (or score) indicating its likelihood of experiencing a landslide, based on the combined influence of the chosen conditioning factors.

# 335 Table 3 Overview of machine learning algorithms for landslide susceptibility analysis

| Algorithm                              | Characteristics                                                                                                                                                                                                                                            | Suitability for Landslide Susceptibility  Mapping                                                                                                                                                                                        | References                                                                        |
|----------------------------------------|------------------------------------------------------------------------------------------------------------------------------------------------------------------------------------------------------------------------------------------------------------|------------------------------------------------------------------------------------------------------------------------------------------------------------------------------------------------------------------------------------------|-----------------------------------------------------------------------------------|
| Logistic<br>Regression (LR)            | - Linear model, relatively simple to implement — Serves as a baseline for comparison more complex models - Interpretable coefficients — Provides insights into the direction magnitude of factor influence data — Easy to calibrate and less prone to over |                                                                                                                                                                                                                                          | (Felicísimo et al.,<br>2013; Matougui<br>and Zouidi, 2025;<br>Zehra et al., 2024) |
| Random Forest<br>(RF)                  | <ul> <li>Ensemble of decision trees</li> <li>Robust to noise and outliers</li> <li>Captures nonlinear relationships</li> <li>Handles high-dimensional data</li> <li>well</li> </ul>                                                                        | <ul> <li>Good balance between interpretability and predictive power</li> <li>Handles mixed data types without extensive preprocessing</li> <li>Resistant to overfitting through bagging</li> </ul>                                       | (Breiman, 2001;<br>Matougui et al.,<br>2023; Tanyu et al.,<br>2021)               |
| XGBoost and<br>LightGBM<br>(XGB & LGB) | <ul> <li>Gradient boosting framework using decision trees</li> <li>High accuracy</li> <li>Effective handling of missing values and complex interactions</li> </ul>                                                                                         | ision trees methods on structured data  - Provides flexible regularisation and fine- tuning options                                                                                                                                      |                                                                                   |
| Multi-Layer Perceptron (MLP)           | <ul> <li>Feedforward neural network</li> <li>Able to learn complex, nonlinear patterns</li> <li>Can incorporate multiple hidden layers for greater capacity</li> </ul>                                                                                     | <ul> <li>Can capture subtle interactions between variables</li> <li>Potential for high predictive performance with proper tuning</li> <li>Complements tree-based methods by leveraging a fundamentally different architecture</li> </ul> | (Gardner and Dorling, 1998; Pham et al., 2017, 2019)                              |

**Table 4 Performance Metrics Used in Model Evaluation** 

| Metric   | Definition                                                                                                           | Formula                                                                                            | Interpretation                                                                                                                                           |
|----------|----------------------------------------------------------------------------------------------------------------------|----------------------------------------------------------------------------------------------------|----------------------------------------------------------------------------------------------------------------------------------------------------------|
| Accuracy | Proportion of correctly classified instances (both landslides and non-landslides).                                   | $\frac{\mathrm{TP} + \mathrm{TN}}{\mathrm{TP} + \mathrm{TN} + \mathrm{FP} + \mathrm{FN}}$          | Measures overall correctness.  High accuracy suggests that the model is generally effective, but may mask class imbalance issues.                        |
| F1-Score | Harmonic mean of Precision and Recall.                                                                               | 2 * Pecision * Recall<br>Recall + Precision                                                        | Balances Precision and Recall. A high F1 indicates strong performance in correctly identifying landslides while minimising false positives or negatives. |
| ROC AUC  | Area Under the Receiver Operating Characteristic Curve, indicating how well the model separates positives/negatives. | Area under the curve depicting the true positive rate (TPR) against the false positive rate (FPR). | Ranges from 0.5 (chance) to 1.0 (perfect discrimination). Higher values indicate stronger overall classification performance.                            |

Figure 5 Flowchart of the methodology adopted

Figure 6 Landslide inventory and stable areas used to implement the landslide susceptibility models (© Esri, © Open Street Map contributors, and the GIS user community, Source: Esri, Maxar, Earthstar Geographics, and the GIS User Community)

#### 3. Results and discussion




# 3.1. Multicollinearity Assessment

To ensure the reliability of model coefficients and predictions, potential multicollinearity among the landslide conditioning factors was examined using the Variance Inflation Factor (VIF). This statistic quantifies how much the variance of a given regression coefficient is inflated due to correlations with other predictors (Kyriazos and Poga, 2023). In general, a VIF value above 5 indicates moderate multicollinearity, whereas values exceeding 10 suggest a more serious concern that may distort model estimates (O'Brien, 2007).

In **Fig. 7**, two bar plots present VIF values for the variables under different threshold considerations. For the factors considered, Elevation displays a notably high VIF (>40), implying a strong correlation with other factors, likely because elevation is closely tied to terrain attributes such as slope, TWI and stream. Other variables, including Landuse, TWI, and NDVI, exhibit moderate VIF values near or above 10, suggesting some overlap in how moisture, vegetation, and terrain patterns are captured. By contrast, factors such as Curvature, Distance to roads, and Aspect remain below the threshold, indicating relatively low redundancy with other predictors. For the factors selected, which applies a more conservative VIF threshold of 5, a narrower set of factors is highlighted. Once again, Elevation, Landuse, TWI, NDVI, and Lithology are shown to correlate strongly with

each other or with the broader suite of predictors. These findings underscore the importance of caution when selecting variables, since high multicollinearity can degrade predictive performance.

Figure 7 VIF analysis of landslide conditioning factors

#### 3.2. Hyperparameters calibration




In this study, we employed several machine learning algorithms to generate landslide susceptibility models for the Mixed, Urban, and Non-Urban datasets. Before the final evaluation, each algorithm underwent a systematic configuration process, using a Bayesian Optimisation approach (Frazier, 2018). The Bayesian method was preferred over more conventional grid or random search methods due to its efficiency in exploring large, multi-dimensional hyperparameter spaces.

Table 5 shows that some hyperparameters (e.g. the learning rate for XGBoost and LightGBM) remain consistent across the Mixed, Urban, and Non-Urban datasets, indicating robust settings regardless of data composition. Other parameters, such as max\_features and min\_samples\_split in RF, vary considerably, suggesting that each environment demands tailored configurations to capture distinct triggers of slope failure. For instance, LightGBM requires a lower num\_leaves in the Urban dataset, while Random Forest benefits from a higher number of trees in that same setting. Meanwhile, MLP demonstrates increasing alpha in non-urban areas, suggesting that stronger constraint is needed to handle more naturally driven landslides. These results underscore the importance of custom tuning for each model—dataset combination to achieve optimal performance.

#### Table 5 Hyperparameters for each algorithm dataset

|               | Dataset            | Mixed  | Urban  | Non-Urban | Range         |
|---------------|--------------------|--------|--------|-----------|---------------|
|               | max_depth          | 6      | 6      | 6         | (2, 6)        |
| Random Forest | max_features       | 1      | 0.3    | 0.3       | (0.3, 1.0)    |
| Kandom Porest | min_samples_split  | 6      | 2      | 3         | (2, 10)       |
|               | n_estimators       | 138    | 500    | 249       | (50,500)      |
|               | colsample_bytree   | 1      | 0.7    | 0.5       | (0.5,1)       |
|               | learning_rate      | 0.3    | 0.3    | 0.3       | (0.01, 0.3)   |
| XGBoost       | max_depth          | 6      | 5      | 6         | (2, 6)        |
|               | n_estimators       | 234    | 218    | 270       | (50,300)      |
|               | subsample          | 1      | 0.5    | 1         | (0.5, 1.0)    |
|               | bagging_fraction   | 1      | 0.65   | 1         | (0.5, 1.0)    |
|               | bagging_freq       | 1      | 3.5    | 5         | (1, 5)        |
| LightGBM      | feature_fraction   | 0.5    | 0.5    | 0.8       | (0.5, 1.0)    |
| LightODW      | learning_rate      | 0.3    | 0.3    | 0.3       | (0.01, 0.3)   |
|               | n_estimators       | 467    | 425    | 431       | (50, 500)     |
|               | num_leaves         | 36     | 14     | 35        | (2, 40)       |
|               | alpha              | 0.0001 | 0.0059 | 0.032     | (0.0001, 0.1) |
| MLP           | hidden_layer_sizes | 43     | 44     | 44        | (5, 50)       |
|               | learning_rate_init | 0.001  | 0.0285 | 0.0145    | (0.001, 0.1)  |

## 3.3. Landslide susceptibility maps




Fig. 8 presents the landslide susceptibility maps generated under each modelling scenario (Mixt, Non-urban, and Urban) for the five different algorithms. The Landslide Susceptibility Index (LSI) expresses the relative spatial probability of landslide occurrence. It reflects how prone an area is to landslides, without reference to the timing or potential impacts. In this study, LSI values were obtained from the machine learning model outputs, where each grid cell was assigned, a continuous score indicating its relative susceptibility. Higher LSI values correspond to greater likelihood of landslide occurrence. For the Mixed and Non-urban scenarios, the maps display broadly similar spatial patterns in high-susceptibility zones, particularly for the tree-based models and Logistic Regression. Nonetheless, minor variations in the extent and intensity of these zones indicate that the Mixed inventory can subtly alter the models' learned relationships.

In contrast, the Urban maps generally display fewer red patches. This reduction is likely due to the limited combinations of factors used during model training, leading to an underestimation of landslide susceptibility when generalized across the study area. Conversely, the non-urban maps tend to feature larger, more continuous patches of high susceptibility. Among the

algorithms, LightGBM, XGBoost, and Random Forest produce consistent spatial patterns with sharper transitions between high- and low-susceptibility regions. This is attributable to their tree-based structures, which effectively capture nonlinear relationships. In comparison, MLP and Logistic Regression yield smoother, more diffuse transitions between susceptibility levels. Notably, Logistic Regression delineates larger continuous zones at the red-blue interface, likely a consequence of its linear decision boundaries.




Despite these differences, all models consistently highlight similar topographic or geological hotspots prone to landslide failure, although the precise shape and intensity of these hotspots vary slightly among the algorithms. This consistency suggests a common underlying signal, albeit one that is interpreted differently depending on the chosen algorithm. Overall, the observed differences underscore that the choice of algorithm can lead to distinct susceptibility patterns, which may have important implications for risk mapping and resource allocation.

The resulting susceptibility maps not only pinpoint areas that appear obviously prone to slope failure, but also reveal zones of precarious stability that may seem stable at present. Such slopes are highly sensitive to unexamined or unregulated human activities, such as excavation or poorly managed water diversion, which could readily tip them into active instability.

Figure 8 Landslide susceptibility maps using the various models and the different datasets

Figure 9 presents the density distributions of LSI values for landslide and stable cells across the different models and datasets,
with density values computed according to Eq. (1). A clear separation between the two distributions reflects strong model
discrimination, whereas substantial overlap indicates weaker predictive performance.

LightGBM. Urban landslides form a tight mode near 0.8–1.0, while urban non-landslide cells concentrate at low LSI, indicating good discrimination. Non-urban landslides also peak at high LSI but with a broader spread, and the non-urban stable curve shows a long high-LSI tail, implying more false positives. The mixed curves closely track the non-urban shapes, suggesting that, in the combined dataset, non-urban signatures dominate, which dilutes urban landslides.




XGBoost. Among all models, XGBoost shows the clearest discrimination. In the urban dataset, landslide cells cluster sharply at LSI  $\approx 0.95$ , while non-landslide curves collapse toward 0–0.2 with negligible right tails. In non-urban terrain, separation remains strong and superior to LightGBM, with only a small fraction of stable cells assigned high LSI. By contrast, the mixed dataset exhibits a broader, flatter spread of LSIs, reflecting the heterogeneity of urban and non-urban signatures and the resulting dilution of the decision boundary. This reinforces the benefit of modelling the two environments separately.

Random Forest. Urban landslides peak around 0.8 with low urban stable densities at high LSI, evidencing good urban performance. In non-urban areas, landslide densities shift to 0.6–0.8 and overlap more with stable cells, reflecting misclassifications. The mixed curves closely mirror the non-urban shapes, indicating that RF's piecewise partitions are dominated by the more non-urban conditions, which reduces selectivity when both environments are pooled. Environment-specific calibration (or thresholds) would likely improve non-urban specificity.

MLP. The MLP concentrates landslide probabilities at the high end, indicating good sensitivity. Non-landslide curves are mostly compressed below 0.2, yet they retain residual right-tails, more visible for mixed and urban sets, so a small fraction of stable cells is assigned high LSI (false positives). This pattern is consistent with a high-capacity model capturing non-linear interactions but becoming over-confident under heterogeneous predictors.

Logistic Regression. Distributions are broader with substantial overlap, but a consistent right-shift of landslide curves remains: landslide modes lie around 0.60–0.75, while non-landslide modes are closer to 0.20–0.45. The density is highest in urban dataset but shifted to low LSI and it weakens in non-urban and mixed datasets, indicating the poorest discrimination capabilities.

Figure 9 Density distributions of Landslide Susceptibility Index (LSI) values obtained by superposing the landslide inventory with the susceptibility maps produced by five machine learning models

## 3.4. Performance evaluation




From Fig. 10, interesting trend emerges: the urban dataset achieves the highest overall performance despite being the smallest dataset. This is somewhat unexpected, as a limited sample size typically constrains model accuracy. Conversely, the non-urban dataset exhibits the lowest performance across several metrics, which is surprising given that landslides in rural areas are generally assumed to follow more consistent, terrain-driven failure mechanisms. While topographical, geological, and vegetation-related features remain key predictors, the models struggle to distinguish landslide-prone areas as clearly as in urban environments. This suggests that natural slope failure processes may be more complex or influenced by subtle, less directly measurable factors.

Several explanations may account for this counterintuitive result. In urban areas, landslides often create a sharper contrast between unstable and stable cells. As shown in **Fig. 4**, urban landslides occupy narrower, more distinctive ranges in several predictors, whereas non-urban landslides span broader, overlapping ranges with the stable class. This improves model separability despite the smaller sample size. In addition, negative sample selection biases may further accentuate the contrast as stable areas were identified through knowledge-based methods, thereby blurring class distinctions. Finally, urban slopes typically display lower geomorphological complexity and greater uniformity of triggering factors compared with non-urban terrain.

The mixed dataset performs between these two extremes, but its results vary across different metrics. By combining urban and rural characteristics, the dataset benefits from a larger sample size but at the cost of increased heterogeneity, making it harder for models to capture distinct patterns specific to either environment.







When comparing the five algorithms across the datasets (**Fig. 11, Table S1**), a clear difference emerges. The three tree-based methods maintain relatively consistent performance across the different dataset configurations, indicating their robustness in handling a wide range of features and data distributions. However, subtle variations do appear. For instance, LightGBM may take the lead in the non-urban setting, while XGBoost often excels in the Urban dataset. Overall, however, these ensembles exhibit less variability than the other approaches.

By contrast, MLP and Logistic Regression show greater performance swings between the Mixed, Urban, and non-urban datasets. MLP displays notable changes in Precision and Recall, reflecting its sensitivity to differences in dataset size, feature distribution, and hyperparameter settings factors that vary substantially between urban and non-urban landscapes. Logistic Regression experiences the largest performance drop in the non-urban dataset, largely because it relies on linear decision boundaries and struggles with the more complex, non-linear interactions typical of topographically driven environments. This marked variability underscores the importance of selecting models capable of adapting to the interplay of environmental and anthropogenic factors.

Beyond these internal performance metrics, it is also important to situate our findings in relation to previous susceptibility assessments conducted in Constantine Province. Several studies have produced maps using statistical, expert-based, or multi-criteria methods, which provide a useful external reference for comparison with our results.

Landslide susceptibility in Constantine Province has been evaluated in several previous studies using different approaches. For instance, (Achour et al., 2017) analyzed a highway road section using statistical methods; however, their study area does not intersect with ours, limiting the relevance of direct comparison. (Abdı et al., 2021) applied AHP and Fuzzy-AHP methods in a zone that partially overlaps our study area. Although their validation inventory was compiled at a smaller scale, the main landslide-prone zones they identified correspond closely to areas that our mixed and non-urban models classify as high to very high susceptibility. In contrast, their mapping underrepresents small urban landslides, which may explain why our urban model captures additional events not emphasized in their results. Similarly, (Bourenane and Bouhadad, 2021; Bourenane et al., 2015) developed susceptibility maps based on expert judgment and statistical approaches. While their analyses were also conducted at a coarser scale, our non-urban and mixed models broadly agree with their delineation of landslide and highly susceptible areas. Taken together, these consistencies with earlier studies provide an indirect form of external validation, supporting the reliability of our susceptibility models. Despite a smaller study area, this work represents the most comprehensive assessment to date of landslide susceptibility in the Constantine region. It stands out for its spatial scale, the level of detail and reliability of the compiled inventory, the integration of advanced learning methods, and advanced analysis of the findings.

Figure 10 Comparison of model performance across different dataset configurations

Figure 11 Model-wise performance evaluation

## 3.5. Calibration of the models


Calibration plots are essential tools for assessing how well a model's predicted probabilities align with actual outcomes. When predictions of models lie above the diagonal, it indicates an underestimation of the true probability of a positive event. Conversely, predictions below the diagonal suggest an overestimation. Ideally, predictions that fall on or near the diagonal demonstrate strong calibration.

According to **Fig. 12**, the calibration plots for the Urban dataset show that MLP and LightGBM align more closely with the diagonal line of perfect calibration, though minor deviations appear at the lower and upper probability extremes. Random Forest and Logistic Regression exhibit larger discrepancies, especially for higher predicted probabilities, where Logistic Regression underperforms considerably. These patterns suggest that in urban contexts, where anthropogenic factors are particularly influential, the complex algorithms capture probability estimates more consistently, while simpler or more rigid

approaches struggle to match actual landslide occurrence. In the Non-urban setting, MLP, XGBoost and LightGBM maintain reasonable calibration across most probability bins, whereas Random Forest underestimates risk at the lower end and overestimate at higher probabilities. By contrast, Logistic Regression show better calibration compared to Urban dataset, although the underestimation of the phenomenon is significative for high susceptibility.



The Mixed dataset, yields overall better-calibrated predictions. All models are stable, remaining close to the diagonal over a wider range of probabilities, while Random Forest and Logistic Regression show less deviation than in the purely urban or non-urban subsets. This suggests that pooling diverse samples may help average out some extremes and lead to more balanced probability estimates, albeit at the potential cost of diluting context-specific patterns evident in separate urban or rural analyses.

Figure 12 Calibration plots for each model and dataset

## 3.6. Relative importance of the factors for the best model

- According to **Figure 13** In the urban model, Aspect emerges as the most variable predictor, suggesting that slope orientation plays a decisive role in urban landslide susceptibility. This effect likely stems from microclimatic differences, such as sunlight exposure and moisture retention which, when combined with urban construction practices, can either exacerbate or mitigate slope instability. Slope follows closely, reinforcing the idea that steep terrains pose significant risks, even in built-up environments.
- A noteworthy finding is that high distances to streams (i.e., being far from streams) show positive SHAP values, which may appear counterintuitive. In many cities, however, watercourses are often located on flatter, lower-lying land, which might also have received better drainage or protective infrastructure. In contrast, the more intensively built slopes away from streams may lack adequate water management, thereby increasing landslide susceptibility. Areas closer to streams may benefit from slope stabilization measures or less steep terrain, resulting in comparatively lower SHAP values. Regarding lithology, the codes range from:
  - 0 = Alluvium
  - 1 = Limestone





- 2 = Conglomerate
- 3 = Brown clayey marl
- 4 = Ocher-brown marly clay

In urban settings, especially problematic lithologies (3 and 4) may be excavated or heavily engineered, which can reduce their inherent instability reflected in more neutral or negative SHAP values. Conversely, if these weak lithologies remain untreated or poorly managed (e.g adjacent to inadequate construction sites), they tend to produce positive SHAP values, indicating higher landslide risk. Although still important, Distance to roads is not as dominant here as in the non-urban model. This likely reflects the high density of roads in urban areas, where being close to a road may sometimes reduce slope risk (due to engineered supports) or increase it (through cut-and-fill activities). The result is a more "mixed" overall impact on the model. Finally, NDVI (a proxy for vegetation density) is generally low in urban environments. Where NDVI is relatively higher (e.g parks or green slopes), the model assigns negative SHAP values (less risk). Conversely, extremely low NDVI levels (sparse or no vegetation) tend to correlate with positive SHAP values, implying greater susceptibility to slope failure.

For non-urban model, Distance to roads stands out as the primary factor. Even limited road networks in rural or natural areas can have disproportionately large destabilizing effects, such as cutting slopes or altering local drainage. Consequently, points nearer to roads show strongly positive SHAP values, signifying elevated landslide risk, which is in accordance with (He et al., 2024). Distance to stream and Slope also have broad SHAP distributions, demonstrating the importance of traditional geomorphological processes in non-urban areas. Being close to a stream often increases undercutting, erosion, and soil saturation, hence positive SHAP values. Areas far from watercourses typically exhibit negative values, indicating reduced landslide likelihood. Aspect and NDVI follow next in importance. High vegetation cover stabilizes slopes (negative SHAP), whereas areas with sparse vegetation elevated risk. Aspect values facing roughly 200–360° (often westerly aspects) correlate

with higher landslide probabilities, likely reflecting local climate or sunlight conditions. While Lithology and Curvature show narrower ranges of influence, they still matter. In non-urban areas, weaker lithologies (3 and 4) are less likely to be engineered or reinforced, so they tend to yield positive SHAP values. Conversely, sturdier materials (0 = Alluvium, 1 = Limestone, 2 = Conglomerate) appear less prone to failure, especially on slopes with minimal human disturbance.

In the mixed model, Distance to roads again features prominently, suggesting that roads, through slope cuts and drainage changes, remain a robust driver of landslide occurrence. The model shows that *mid-range* values of Distance to roads in particular may increase risk, possibly reflecting zones of suburban expansion or partial engineering. Slope and Distance to stream are also influential, consistent with gravity-driven and water-induced failure mechanisms. Aspect and Lithology display moderate but noticeable effects. The lithology results resemble those in the urban dataset, indicating the model may be influenced by the greater representation of urban environments in the sample or by shared lithological units between urban and rural areas. NDVI continues to reduce landslide probabilities where vegetation is dense (e.g., forested slopes), although its stabilizing role is less pronounced than in purely rural contexts. Curvature shows a narrower range of SHAP values overall, but still distinguishes concave zones from convex slopes. This aligns with the SHAP interpretation, as concave areas (negative curvature / low values) are inherently more prone to landslides due to the concentration of runoff and the facilitation of water infiltration.

Across all three datasets, both natural factors (slope, distance to streams, lithology) and anthropogenic factors (particularly roads) emerge as key landslide predictors, with their relative importance shifting depending on the urban or non-urban context. In urban environments, natural drainage patterns are often disrupted by impervious surfaces and redirected through engineered systems. Areas farther from natural streams may lack adequate subsurface drainage infrastructure, leading to groundwater accumulation and increased pore water pressure, a primary trigger for slope instability. In contrast, non-urban terrains follow more common geomorphological logic, with proximity to streams or steep slopes strongly increasing instability. The mixed dataset blends these trends, underscoring that roads, topography, and hydrological factors are consistently significant across diverse landscapes. By comparing these results, decision-makers can better tailor landslide mitigation strategies, focusing on slope stabilization and drainage management in urban expansions, while prioritizing safe road infrastructure and vegetation conservation in more rural settings.

Figure 13 Shap plot for the relative importance of the factors for the best model

## 4. Conclusions






This study addresses the critical challenge of landslide susceptibility assessment in rapidly urbanizing regions, with a focus on the city of Constantine, Algeria, a region known to be vulnerable to landslides due to its complex morpho-geology, clay-rich soils, and intense anthropogenic activities. By decoupling urban and non-urban landslide mechanisms, the research provides a nuanced understanding of the distinct factors driving slope instability in these contrasting environments. The integration of geospatial data, machine learning algorithms, and advanced analytical techniques enabled the development of robust landslide susceptibility models tailored to urban, non-urban, and mixed landscapes.

In urban dataset, the obtained results seem counter-intuitive, because in urban areas natural factors over-rule the anthropogenic ones in landslide susceptibility assessment. On the other hand, mixed dataset tends to have more similarities to non-urban dataset and demonstrated intermediate performance due to the larger inventoried slides in non-urban dataset compare to the urban one, highlighting the challenges of modelling heterogeneous landscapes but also the potential for more balanced risk assessments.

Complex algorithms such as XGBoost, LightGBM, and Random Forest consistently outperformed simpler models like Logistic Regression MLP across all datasets, particularly for the urban dataset. This superior performance underscores the necessity of employing advanced, non-linear models to accurately assess landslide susceptibility, given the inherent complexity of the phenomenon, which involves intricate interactions between environmental and anthropogenic factors. The calibration of these models further supports this conclusion, as it highlights the non-linear nature of the problem, with complex algorithms demonstrating better alignment between predicted probabilities and actual outcomes. In contrast, the LR model, which does not require fine-tuning, exhibited the opposite trend in performance, consistently underperforming compared to the other models. This stark contrast emphasizes the critical role of hyperparameter optimization and the need for more sophisticated modelling approaches to capture the multifaceted dynamics of landslide susceptibility, particularly in urban settings where human activities significantly influence slope stability.

An important outcome of this work is that the different algorithms not only vary in predictive accuracy but also in their estimation tendencies. Boosting methods achieved the strongest discrimination but tended to underestimate susceptibility in marginally unstable zones, Random Forest produced the most balanced delineation of high-susceptibility areas, while Logistic Regression consistently overestimated susceptibility, mapping the largest hazard zones. This gradient of behaviours suggests that model choice should not be guided by accuracy alone but also by the level of security desired in planning and mitigation. In precautionary contexts, an overestimating model may be preferable, while in operational resource management, a balanced or conservative model may be more appropriate.

The study's methodological framework, which incorporates identification and detailed inventory of landslides, advanced machine learning techniques, interpretable factor analysis an uncoupled inventory, represents a significant advancement in landslide susceptibility mapping. By addressing the limitations of existing studies that often treat urban and non-urban environments as homogeneous, this research provides a more accurate and actionable basis for spatial planning and risk

mitigation. The findings underscore the importance of context-specific approaches to landslide hazard assessment, particularly in regions undergoing rapid urbanization, and offer valuable insights for policymakers, urban planners, and disaster management authorities aiming to enhance community resilience and sustainable development. Moreover, in our study area, the broader regional context is one of precarious geomorphological stability, where anthropogenic interventions can either stabilise or destabilise the terrain, ultimately interacting with natural processes in a highly complex manner.

## **Data Availability Statement**

Some data, models, or code that support the findings of this study are available from the corresponding author upon reasonable request.

# 605 Acknowledgments



**Conflict of interest**: The authors declare that they have no conflict of interest.

Funding: The authors did not receive support from any organization for the submitted work.

#### **Author's contribution:**

Conceptualization, M.Z.; methodology, D.M. and M.Z.; software, M.Z.; validation, D.M.Y., D.M. M.Z. and B.C.; formal analysis, D.M.Y., D.M. and M.Z.; investigation, D.M.Y., D.M. M.Z. and B.C.; data curation, M.Z and D.M.Y.; writing—original draft preparation, M.Z. writing—review and editing, M.Z., D.M.Y., D.M. and B.C.; visualization., M.Z.; supervision, B.C. All authors have read and agreed to the published version of the manuscript.

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
