# Peer review of "Decoupling Urban and Non-Urban Landslides for Susceptibility Mapping in Transitional Landscapes, a Case Study from Transitional Landscapes in Southwestern Constantine, Algeria"

_EGUsphere, 2025_

## Author Response (AR1)

**Reviewer 1**

Dear reviewer,

Thank you for your valuable comments, which have greatly helped improve the clarity and quality of this manuscript. Please find our detailed responses below.

- 1. In general, the text in some figures is very small and difficult to read. Please make it large enough to print and read on A4 paper.
  - ✓ **Response**: Thank you for this suggestion. Figures 1, 2, 4, 10 and 11 have been reformatted to enhance readability on A4 paper.
- 2. P4 Figure 1 Where is the Constantine study area in Figure 1? Does this refer to the rectangular area of approximately 3km x 5km shown in Figure 1?
  - ✓ Response: The study area corresponds to the red rectangular box shown in Figure 1 within Constantine Province. To avoid confusion, we have revised the figure by improving the layout and annotations, making the boundaries of the study area more explicit.
- 3. P4 Figure 1 How is the extent of the landslide interpreted and shown in red in Figure 1? When a slope collapses, sometimes only the collapsed area is interpreted, and sometimes the area where the collapsed soil and sand have deposited is also interpreted in addition to the collapsed area. How is the area interpreted here? In my opinion, given the main purpose of this paper, the former is preferable.
  - ✓ Response: We thank the reviewer for this pertinent observation. Landslides were delineated by mapping the entire affected area, from the main scarp to the toe, thereby including both the source and accumulation zones. In this region, dominated by clays and marls overlying hard substrates, displacements are moderate and deposition zones relatively limited.
  - ✓ Changes in the manuscript: we added the following paragraph in the Landslide inventory section: "Each landslide was delineated by mapping the entire affected

surface, from the main scarp to the toe, thus incorporating both the source and accumulation zones. Due to the predominance of clays and marls overlying hard limestone or conglomerate formations, most slope movements in the study area are characterized by moderate displacements and relatively small deposition areas."

- 4. P4 L100 Even if I read this section through, it is not clear when the landslide field survey was carried out, meaning that the reader cannot determine when the landslide shown in Figure 1 occurred.
  - ✓ **Response:** We acknowledge the importance of clarifying the inventory timeframe. Field surveys were conducted between June and December 2024.
  - ✓ Changes in the manuscript: This information has been added to the revised manuscript as "The inventory including urban and non-urban landslides were compiled during a comprehensive field and remote sensing interpretation survey conducted between June and December 2024."
- 5. P5 L127 I understand the way how you made landslide inventory in Urbanized area. However, I don't know the method in which you made landslide inventory in non-urbanized area. Would you tell it to me and the readers? At first, I tried to find the method in Alharbi et al(2014), but I failed. The literature is just describing rural slope failures in Faifa in Saudi Arabia.
  - ✓ Response: We agree that the description of the inventory mapping method for non-urbanized areas was insufficiently detailed. In non-urbanized sectors of the study area, landslides were primarily identified using remote sensing techniques, supported by field verification. The landslide inventory was established based on morphological criteria, following the principles proposed by (Varnes, 1984). This approach relies on the visual interpretation of geomorphic features typically associated with mass movements, such as main scarps, toe bulges, surface cracks, and accumulation zones. These morphodynamic indicators make it possible to identify and delineate landslides. Therefore, the method applied in rural settings is essentially based on direct geomorphological analysis, which complements the approach adopted in urbanized areas.

- ✓ Changes in the manuscript: We have added a sub section 'Landslide Inventory in Non-Urbanized Areas: In non-urbanized areas, the landslide inventory was constructed primarily through remote sensing interpretation, complemented by selective field verification. Mapping was guided by morphological criteria following the principles of (Varnes, 1984) whereby typical geomorphic signatures of mass movements, such as scarps, displaced material, surface cracks, and toe bulges, were visually identified and delineated. This geomorphological approach ensured that landslides in rural sectors were consistently captured and provided a methodological counterpart to the field-based strategy applied in urban areas.'
- 6. P6L144 Fig.4 b-f ---> Fig.2 b&f? To me "b-f" seems to indicate from b to f, that is "b, c, d, e, f".
  - ✓ **Response:** Thank you for spotting this inconsistency. We have corrected the figure references in the revised manuscript.
- 7. P8 Table 1 Was the NDVI calculated based on the imagery on 28 March 2017? Or that on 2 February 2025? Why did you use two images? I think that readers may be wondering which came first: the dates the satellite images were taken or the dates the landslides occurred.
  - ✓ **Response:** We thank the reviewer for highlighting this ambiguity. NDVI was derived from a multi-temporal series of harmonized Sentinel-2 MSI images spanning 28 March 2017 to 2 February 2025, rather than a single scene. This approach reduces cloud contamination, seasonal effects, and signal noise. We used the median NDVI across the time series.
  - ✓ Changes in the manuscript: This has been clarified in Table 1 Summary of landslide conditioning factors used in this study.

| Factor Type | Source | Resolution |
|-------------|--------|------------|
|             |        |            |

| NDVI | Harmonised Sentinel-2 MSI (median of time series between 28 March 2017 and 2 February 2025) (Claverie et al., 2018) | 10 etres |
|------|---------------------------------------------------------------------------------------------------------------------|----------|
|      |                                                                                                                     |          |

- 8. P10 L199 Slope < 5 degree ---> slope > 5 degree?
  - ✓ **Response:** We thank the reviewer for pointing out this ambiguity. Our intention was to indicate that landslide occurrence increases with steeper slope angles. The threshold value in the text was a typographical oversight. The correct statement should read "slope > 5°" (not "slope < 5°"), since in our study area slopes above this threshold showed markedly higher landslide frequencies.
  - ✓ Changes in the manuscript: Certain factor ranges (e.g., slope >5° or elevation between 500 and 600 m) align with higher or lower landslide frequencies, informing both susceptibility modelling and mitigation strategies.
- 9. P10 L201 What this sentence shows strongly depends on the definition of the "urban landslide area". What is the urban landslide area?
  - ✓ Response: We thank the reviewer for this observation. The definition of the "urban landslide area" is provided in the manuscript under the subsection Landslide Inventory in Urbanized Areas. "To establish a comprehensive inventory within urbanized portions of the study area, landslide identification was primarily based on in-situ observations."
  - ✓ Changes in the manuscript: For more clarity we edited this part as: "To establish a comprehensive inventory within urbanized portions of the study area, landslide identification was primarily based on in-situ observations and mapped where polygons intersect with built-up zones, as defined by the land cover map and validated through local land use."
- 10. P11 L225 At what angle does a slope have to be considered "steep" or "low"? According to Figure 4, the difference in occurrence between urban and non-urban landslides appears

to be the difference in landslide occurrence density on slopes of 10 degrees or steeper. So, do you call slopes of 10 degrees or more "steeper slopes"?

- **Response**: In this study, we identified an empirical threshold of about  $8^{\circ}$ , as shown in Figure 4. At this point, the non-urban landslide density begins to exceed that of non-landslide terrain, providing a natural break between "moderate" and "steeper" slopes. Accordingly, we refer to gentle/low slopes as  $<8^{\circ}$  and steeper slopes as  $\ge 8^{\circ}$ . While non-urban landslides are concentrated on  $\ge 8^{\circ}$ , urban landslides also occur on slopes  $<8^{\circ}$  due to anthropogenic disturbances such as excavation and drainage mismanagement.
- ✓ Changes in the manuscript: In non-urban regions, landslides predominantly occur on steeper slopes (≥8°), where gravitational failures are more frequent in the absence of human disturbance. In urban settings, landslides are common on moderate to steep slopes but may also develop on gentle slopes (<8°) when construction activities when construction practices, such as excavation and drainage mismanagement, undermine natural stability. Although slope remains a primary driver of landslides across both contexts, urban activities can widen the range of vulnerable gradients.

**11. P11 L240 - This sentence seems to be difficult to understand**

- ✓ **Response:** We thank the reviewer for pointing this out. The sentence has been revised for clarity.
- ✓ Changes in the manuscript: "From the density curves, urban landslides are generally associated with lower NDVI values, reflecting the reduced vegetation cover typical of built-up environments. In contrast, non-urban landslides often occur at moderately higher NDVI levels, where vegetation provides low root reinforcement. Nevertheless, agricultural and semi-natural areas may still experience slope failures when land management practices such as deforestation or inadequate irrigation degrade vegetation quality. This pattern indicates that vegetation cover alone does not guarantee slope stability"

- 12. P12 Figure 4 I think that the definition of the landslide density should be obviously shown using an equation if possible
  - ✓ **Response:** We thank the reviewer for this helpful suggestion. The densities shown in Figure 4 were obtained using a Kernel Density Estimation (KDE) approach, which provides a smoothed representation of the probability distribution of landslide versus non-landslide cells for each conditioning factor.
  - ✓ Changes in the manuscript: To improve clarity, we have now added the KDE formulation in the "Landslide causative factors" section as follows:

To understand the influence of environmental and anthropogenic factors on slope stability, probability density functions were estimated for each conditioning variable using a kernel density estimator (KDE):

$$\hat{f}_h(x) = \frac{1}{nh} \sum_{i=1}^n K\left(\frac{x - x_i}{h}\right)$$

where  $\hat{f}_h(x)$ : estimated probability density at x, n the number of observations, h the bandwidth (smoothing parameter), K the kernel function (e.g., Gaussian), and  $x_i$  the individual observations.

This approach provides a smoothed representation of the distributions of landslide and non-landslide cells, while allowing urban and non-urban landslide occurrences to be analyzed separately.

- 13. Rural landslide ---> Non-urban landslide? Rural? Non-urban? Are they different from each other?
  - Response: We thank the reviewer for highlighting this inconsistency. In our study, "rural" and "non-urban" were intended to refer to the same category of landslides occurring outside built-up areas. To avoid confusion, we have standardized the terminology throughout the manuscript and now consistently use the term "non-urban landslides." This ensures clear distinction between urban (within built-up areas) and non-urban (outside built-up areas) contexts.

- 14. P13 L26 In the latter sentences, you mentioned "...the urban dataset achieves the highest overall performance despite being the smallest dataset". To allow readers to find that the urban dataset is the smallest, you should show some evidence somehow. How about adding one table to show the number and the area of landslides for each subset, Urban, Non-urban, and Mixed? The maximum, minimum, and average size of landslides for each subset should be also shown. It might be helpful for readers to understand the landslide characteristics.
  - ✓ **Response:** We thank the reviewer for this comment. To provide clear evidence that the urban dataset is the smallest, we computed descriptive statistics for the urban, non-urban, and mixed landslide inventories.
  - ✓ Changes in the manuscript: To further characterize the mapped landslides, descriptive statistics were calculated for the urban, non-urban, and mixed inventories. The urban dataset comprises 123 landslides totaling 18.4 ha, while the non-urban dataset includes 61 landslides covering 21.2 ha. Combined, the mixed inventory contains 184 landslides with a total area of 39.6 ha. Despite its larger number of events, the urban dataset represents the smallest total area.

Table 1. Descriptive statistics of mapped landslides in the study area

| Туре      | Count | Total
Area (ha) | Mean Area (ha) | Median Area (ha) | Min Area (ha) | Max
Area (ha) |
|-----------|-------|--------------------|----------------|------------------|---------------|------------------|
| Non-urban | 61    | 21.24              | 0.348          | 0.111            | 0.0058        | 3.97             |
| Urban     | 123   | 18.4               | 0.1496         | 0.041            | 0.0006        | 4.37             |
| Mixed     | 184   | 39.64              | 0.215          | 0.051            | 0.0006        | 4.37             |

- 15. P13 L275 Finally, there appears to be no explanation of how to classify landslides into urban, non-urban, and mixed datasets.
  - ✓ **Response:** We thank the reviewer for pointing out this missing clarification. In the revised manuscript, we now provide a clear description of the procedure used to classify landslides into urban, non-urban, and mixed datasets. Landslides were first delineated as polygons and overlaid with the official land-cover dataset. An event was classified as *urban* when its polygon intersected built-up zones or was directly associated with infrastructures. Conversely, events located entirely outside these zones, within bare or agricultural land cover, were classified as *non-urban*.
  - ✓ Changes in the manuscript: The following paragraph were added to the Landslide inventory section: "Landslide inventory classification: The landslide inventory was constructed with the objective of distinguishing between urban and non-urban slope failures while maintaining methodological consistency across the study area. Classification was carried out by overlaying mapped landslide polygons with the land-cover dataset. An event was defined as urban when its polygon intersected built-up zones or close to infrastructure. Conversely, landslides located entirely outside these zones and in areas characterized by bare or agricultural land cover were classified as non-urban."

**16. P17 L 311 - VIF (>30) ---> VIF (>40)?**

- ✓ **Response:** We thank the reviewer for noticing this inconsistency. The correct threshold used in our analysis is **VIF** > **40**, not 30. We have corrected this in the revised manuscript.
- 17. P19 L336 You should describe the definition of the landslide susceptibility index shown in Figure 8.
  - ✓ **Response:** We have now clarified the definition of the Landslide Susceptibility Index (LSI) in the revised manuscript. LSI represents the relative spatial probability that a given location is prone to landslides, based on environmental, geological, and anthropogenic conditioning factors. Unlike hazard or risk,

susceptibility refers only to predisposition and does not include a time component or expected consequences. In our study, the LSI was derived from the machine learning model outputs, where each grid cell was assigned, a continuous value reflecting its relative likelihood of landslide occurrence. Higher values correspond to areas of greater susceptibility.

- ✓ Changes in the manuscript: "The Landslide Susceptibility Index (LSI) expresses the relative spatial probability of landslide occurrence. It reflects how prone an area is to landslides, without reference to the timing or potential impacts. In this study, LSI values were obtained from the machine learning model outputs, where each grid cell was assigned, a continuous score indicating its relative susceptibility. Higher LSI values correspond to greater likelihood of landslide occurrence."
- 18. In this section, landslide susceptibility maps using the various models and the different datasets are only compared to each other. This is important, but I think there is one thing missing. That is these maps should be also compared to the real landslide inventory shown in Figure 1. Furthermore, not only landslide inventory but also the "stable areas" shown in Figure 6 should be compared to the landslide susceptibility maps. Some stable areas seem to be evaluated highly susceptible in some models.
  - Response: We thank the reviewer for this valuable remark. In order to compare the maps with each other, we propose to analyse the superposition of landslide and stable areas on the susceptibility maps, and to display this comparison through their distribution of susceptibility values. This approach, illustrated in Figure 9, highlights how each model differentiates landslide cells from stable cells: landslides should cluster at higher susceptibility values, while stable areas should be concentrated at lower values. This complementary analysis allows us to directly assess model performance with respect to both the landslide inventory and stable areas.

✓ Changes in the manuscript: "Figure 9 compares the density distributions of Landslide Susceptibility Index (LSI) values for landslide cells and stable cells across models and datasets. Clear separation between the two distributions indicates good discrimination.

LightGBM. Urban landslides form a tight mode near 0.8–1.0, while urban non-landslide cells concentrate at low LSI, indicating good discrimination. Non-urban landslides also peak at high LSI but with a broader spread, and the non-urban stable curve shows a long high-LSI tail, implying more false positives. The mixed curves closely track the non-urban shapes, suggesting that, in the combined dataset, non-urban signatures dominate, which dilutes urban landslides.

XGBoost. Among all models, XGBoost shows the clearest discrimination. In the urban dataset, landslide cells cluster sharply at LSI  $\approx 0.95$ , while nonlandslide curves collapse toward 0–0.2 with negligible right tails. In nonurban terrain, separation remains strong and superior to LightGBM, with only a small fraction of stable cells assigned high LSI. By contrast, the mixed dataset exhibits a broader, flatter spread of LSIs, reflecting the heterogeneity

of urban and non-urban signatures and the resulting dilution of the decision boundary. This reinforces the benefit of modelling the two environments separately.

Random Forest. Urban landslides peak around 0.8 with low urban stable densities at high LSI, evidencing good urban performance. In non-urban areas, landslide densities shift to 0.6–0.8 and overlap more with stable cells, reflecting misclassifications. The mixed curves closely mirror the non-urban shapes, indicating that RF's piecewise partitions are dominated by the more non-urban conditions, which reduces selectivity when both environments are pooled. Environment-specific calibration (or thresholds) would likely improve non-urban specificity.

MLP. The MLP concentrates landslide probabilities at the high end, indicating good sensitivity. Non-landslide curves are mostly compressed below 0.2, yet they retain residual right-tails, more visible for mixed and urban sets, so a small fraction of stable cells is assigned high LSI (false positives). This pattern is consistent with a high-capacity model capturing non-linear interactions but becoming over-confident under heterogeneous predictors.

Logistic Regression. Distributions are broader with substantial overlap, but a consistent right-shift of landslide curves remains: landslide modes lie around 0.60–0.75, while non-landslide modes are closer to 0.20–0.45. The density is highest in urban dataset but shifted to low LSI and it weakens in non-urban and mixed datasets, indicating the poorest discrimination capabilities."

**References:**

Claverie, M., Ju, J., Masek, J. G., Dungan, J. L., Vermote, E. F., Roger, J. C., Skakun, S. V., and Justice, C.: The Harmonized Landsat and Sentinel-2 surface reflectance data set, Remote Sensing of Environment, 219, 145–161, https://doi.org/10.1016/j.rse.2018.09.002, 2018.

Varnes, D. J.: Landslide hazard zonation: a review of principles and practice, 1984.

**Reviewer 2**

Dear reviewer,

We thank you for your constructive comments, which have helped improve the clarity, depth, and overall quality of our manuscript. Please find the detailed responses below.

- 19. While the decoupling of urban and non-urban landslides is the main contribution, the introduction and discussion do not sufficiently contrast this approach with existing international studies. Please highlight more explicitly what gap in the literature is addressed and how this study advances current knowledge.
  - Response: We thank the reviewer for this valuable comment. In the revised manuscript, we strengthened the Introduction by explicitly contrasting our approach with previous international studies. We now highlight that, although machine learning has been widely applied in landslide susceptibility mapping (LSM), most existing research treats the landscape as a single entity without distinguishing urban from non-urban environments.
  - ✓ Changes in the manuscript: We have improved the discussion about this shortcoming as follow: "Although extensive research has been carried out on landslide hazard assessment in urban areas (Bathrellos et al., 2009; Huang et al., 2023; Pascale et al., 2010, 2013), these studies generally examine urban environments in isolation. They rarely investigate how urban and non-urban processes differ, particularly in transitional zones where the two settings interact. At the same time, the broader international literature shows that machine learning (ML) has become a widely adopted tool for landslide susceptibility mapping. However, most applications still treat the landscape as a homogeneous entity, without explicitly distinguishing between urban and non-urban contexts.

Early work illustrates this limitation. For instance, (Caniani et al., 2008) applied artificial neural networks in Potenza, Italy, but without differentiating urban from non-urban landslides either in the inventory or during modeling. More recent studies have followed a similar approach. (Islam et al., 2025) applied a hybrid ML model to a rapidly urbanizing area in Bangladesh, and (Luo et al., 2025) assessed landslide hazards in the central Guizhou urban agglomeration using SVM, DNN, and bagging

algorithms; in both cases, inventories did not explicitly separate urban from nonurban events.

This limitation is critical because urban landslides are often shaped by anthropogenic factors, including slope modification, drainage alteration, construction practices, and infrastructure density, that differ markedly from the natural drivers that dominate in non-urban areas. Without explicitly accounting for these differences, susceptibility models risk oversimplifying complex processes and overlooking the distinct mechanisms of slope failure across contrasting environments."

- 20. The construction of the landslide inventory is described, but the exact criteria for classifying events as "urban" or "non-urban" remain unclear. More detail is needed on how transitional or mixed zones were treated.
  - ✓ **Response**: We agree that clearer detail on classification criteria is necessary. In the revised manuscript, we now explicitly describe the procedure used to differentiate urban from non-urban landslides. An event was classified as urban when its mapped polygon intersected built-up areas or infrastructures (residential blocks, roads, retaining structures, utilities) as defined by the land cover map, while landslides located entirely in natural or agricultural zones were classified as non-urban.
  - ✓ Changes in the manuscript: The following clarification were added to the Landslide inventory section: "Landslide inventory classification: The landslide inventory was constructed with the objective of distinguishing between urban and non-urban slope failures while maintaining methodological consistency across the study area. Classification was carried out by overlaying mapped landslide polygons with the land-cover dataset. An event was defined as urban when its polygon intersected built-up zones or close to infrastructure. Conversely, landslides located entirely outside these zones and in areas characterized by bare or agricultural land cover were classified as non-urban."
- 21. The finding that urban datasets outperform rural datasets is described as "unexpected." This requires deeper explanation—possible reasons include smaller sample size, greater homogeneity of urban triggers, or biases in negative sample selection.

- Response: We agree that the superior results from the urban dataset warrant additional explanation, particularly given its smaller size. Specifically, we now discuss how urban landslides create a sharper contrast between unstable and stable cells, as illustrated in Fig. 4, where urban events occupy narrower and more distinctive ranges of several conditioning factors, while non-urban landslides span broader, overlapping ranges. This enhances model separability despite the smaller sample size. We also acknowledge the potential role of negative sample selection biases, as stable areas were identified through knowledge-based methods, which may blur class distinctions in rural terrain. Finally, we clarify that urban slopes generally present lower geomorphological complexity and more uniform triggering factors compared with rural environments. processes in rural settings. These elements likely ease the classification task for models in urban contexts, despite the limited sample volume.
- Changes in the manuscript: We have improved the discussion as follow: "Several explanations may account for this counterintuitive result. In urban areas, landslides often create a sharper contrast between unstable and stable cells. As shown in Fig. 4, urban landslides occupy narrower, more distinctive ranges in several predictors, whereas non-urban landslides span broader, overlapping ranges with the stable class. This improves model separability despite the smaller sample size. In addition, negative sample selection biases may further accentuate the contrast as stable areas were identified through knowledge-based methods, thereby blurring class distinctions. Finally, urban slopes typically display lower geomorphological complexity and greater uniformity of triggering factors compared with rural terrain."
- 22. SHAP-based interpretations are sometimes vague (e.g., "being farther from streams may increase instability"). More context-specific engineering explanations should be provided.
  - ✓ **Response**: We acknowledge the need for more precise and context-specific interpretations of the SHAP results.
  - ✓ Changes in the manuscript: The improved description now reads: "Across all three datasets, both natural factors (slope, distance to streams, lithology) and anthropogenic factors (particularly roads) emerge as key landslide predictors, with their relative importance shifting depending on the urban or non-urban context. In

urban environments, natural drainage patterns are often disrupted by impervious surfaces and redirected through engineered systems. Areas farther from natural streams may lack adequate subsurface drainage infrastructure, leading to groundwater accumulation and increased pore water pressure, a primary trigger for slope instability. In contrast, non-urban terrains follow more common geomorphological logic, with proximity to streams or steep slopes strongly increasing instability. The mixed dataset blends these trends, underscoring that roads, topography, and hydrological factors are consistently significant across diverse landscapes. By comparing these results, decision-makers can better tailor landslide mitigation strategies, focusing on slope stabilization and drainage management in urban expansions, while prioritizing safe road infrastructure and vegetation conservation in more rural settings."

- 23. Validation relies solely on cross-validation and internal metrics. If feasible, please add independent validation (e.g., comparison with external maps or independent inventory) or at least discuss this limitation.
  - Response: We agree that external, independent validation offers a more comprehensive assessment of a model's true predictive power. However, the available landslide inventory for the study area is relatively limited in size and has already been partitioned into separate urban and non-urban witch reduce the data even more. This subdivision limits the possibility of extracting a completely independent inventory, making the option of additional validation less suitable in the context of this work. We also appreciate the reviewer's suggestion regarding comparison, and we have added a discussion contrasting our results with existing susceptibility maps from previous studies in Constantine Province. This provides an additional, indirect form of external validation.
  - ✓ **Changes in the manuscript:** We have added the following discussion in the results and discussion section:

"Beyond these internal performance metrics, it is also important to situate our findings in relation to previous susceptibility assessments conducted in Constantine Province. Several studies have produced maps using statistical, expert-based, or multi-criteria methods, which provide a useful external reference for comparison with our results.

Landslide susceptibility in Constantine Province has been evaluated in several previous studies using different approaches. For instance, (Achour et al., 2017) analyzed a highway road section using statistical methods; however, their study area does not intersect with ours, limiting the relevance of direct comparison. (Abdı et al., 2021) applied AHP and Fuzzy-AHP methods in a zone that partially overlaps our study area. Although their validation inventory was compiled at a smaller scale, the main landslide-prone zones they identified correspond closely to areas that our mixed and non-urban models classify as high to very high susceptibility. In contrast, their mapping underrepresents small urban landslides, which may explain why our urban model captures additional events not emphasized in their results. Similarly, (Bourenane and Bouhadad, 2021; Bourenane et al., 2015) developed susceptibility maps based on expert judgment and statistical approaches. While their analyses were also conducted at a coarser scale, our non-urban and mixed models broadly agree with their delineation of landslide and highly susceptible areas. Despite a smaller study area, this work represents the most comprehensive assessment to date of landslide susceptibility in the Constantine region. It stands out for its spatial scale, the level of detail and reliability of the compiled inventory, the integration of advanced learning methods, and advanced analysis of the findings."

- 24. Discussion should go beyond performance ranking to highlight the strengths, weaknesses, and applicability of each algorithm.
  - Response: We thank the reviewer for this constructive suggestion. In the revised manuscript, we expanded the Results and discussion section to provide a detailed evaluation of the strengths, weaknesses, and applicability of each algorithm. Specifically, we explain that boosting methods (XGBoost and LightGBM) achieved the highest predictive performance but tended to underestimate susceptibility zones, yielding more conservative predictions. Random Forest, while slightly less accurate in raw metrics, offered the best balance between over- and underestimation, making it a robust and pragmatic choice for operational applications. The Multi-Layer Perceptron showed capacity to capture complex non-linear interactions but displayed variable performance in smaller or subdivided datasets, suggesting it is better suited to larger or multi-temporal inventories. Logistic Regression, although

the least accurate, systematically overestimated susceptibility and delineated the largest high-risk zones, which may be advantageous in contexts where maximum precaution is required.

✓ **Changes in the manuscript:** We have added the following subsection to the manuscript: "Strengths, limitations of the algorithms

While performance metrics provide a quantitative comparison of the models, it is equally important to examine their qualitative strengths, limitations, and practical applicability. Each algorithm interprets the data in a different way, leading to distinctive patterns of susceptibility mapping, ranging from conservative underestimation to precautionary overestimation.

XGBoost and LightGBM consistently achieved the highest predictive performance in this study. Their strength lies in their ability to capture non-linear interactions between conditioning variables and to partition the feature space into highly discriminative regions. This capacity was particularly evident in the urban dataset, where landslide and non-landslide cells display sharp contrasts in predictor ranges. Both boosting models also incorporate advanced regularization, which helps to prevent overfitting in relatively small samples. However, one limitation observed is a tendency to underestimate susceptibility in certain marginal areas, leading to smaller zones classified as highly susceptible compared to other algorithms and expert insights. This suggests that, while boosting methods maximize accuracy, they may provide conservative predictions that require careful interpretation in risk-averse contexts.

Random Forest also demonstrated strong and stable performance across datasets. Although slightly less precise than boosting methods in terms of raw metrics, it offered the best overall balance between overestimation and underestimation of susceptibility zones. Its robustness to noise and low sensitivity to hyperparameter settings make it an attractive choice for operational applications, particularly where inventories are small or unevenly distributed. Random Forest also proved effective in identifying consistent susceptibility patterns in both urban and non-urban settings, highlighting its reliability as a middle-ground solution that balances predictive strength with practical usability.

The Multi-Layer Perceptron, by contrast, showed more variable performance. It was capable of capturing complex, non-linear patterns and sometimes rivalled tree-based methods in predictive accuracy, but its sensitivity to dataset and tuning was evident.

With smaller inventories, particularly after splitting into urban and non-urban subsets, MLP became less reliable (High spread of performance metrics) and producing higher false positives. As such, MLP is better suited to larger or to contexts where multi-temporal data are available to stabilize training.

Logistic Regression served as a valuable baseline model, offering transparency and straightforward interpretability of predictor effects. However, the simplicity of Logistic Regression also represents its main limitation: it relies on linear decision boundaries and systematically overestimated susceptibility in our study, producing the largest areas classified as high or very high risk. While this may reduce precision, it could also be advantageous in contexts where a high level of precaution is required, since it minimizes the risk of overlooking unstable zones. Logistic Regression thus remains valuable for rapid preliminary assessments, for communicating clear risk signals, and in situations where maximal safety margins are prioritized."

**References:**

Abdı, A., Bouamrane, A., Karech, T., Dahri, N., and Kaouachi, A.: Landslide Susceptibility Mapping Using GIS-based Fuzzy Logic and the Analytical Hierarchical Processes Approach: A Case Study in Constantine (North-East Algeria), Geotechnical and Geological Engineering, 39, 5675–5691, https://doi.org/10.1007/s10706-021-01855-3, 2021.

Achour, Y., Boumezbeur, A., Hadji, R., Chouabbi, A., Cavaleiro, V., and Bendaoud, E. A.: Landslide susceptibility mapping using analytic hierarchy process and information value methods along a highway road section in Constantine, Algeria, Arabian Journal of Geosciences, 10, https://doi.org/10.1007/s12517-017-2980-6, 2017.

Bathrellos, G. D., Kalivas, D. P., and Skilodimou, H. D.: GIS-based landslide susceptibility mapping models applied to natural and urban planning in Trikala, central Greece, Estudios Geologicos, 65, 49–65, https://doi.org/10.3989/egeol.08642.036, 2009.

Bourenane, H. and Bouhadad, Y.: Impact of Land use Changes on Landslides Occurrence in Urban Area: The Case of the Constantine City (NE Algeria), Geotechnical and Geological Engineering, 39, https://doi.org/10.1007/s10706-021-01768-1, 2021.

Bourenane, H., Bouhadad, Y., Guettouche, M. S., and Braham, M.: GIS-based landslide susceptibility zonation using bivariate statistical and expert approaches in the city of

Constantine (Northeast Algeria), Bulletin of Engineering Geology and the Environment, 74, 337–355, https://doi.org/10.1007/s10064-014-0616-6, 2015.

Caniani, D., Pascale, S., Sdao, F., and Sole, A.: Neural networks and landslide susceptibility: A case study of the urban area of Potenza, Natural Hazards, 45, 55–72, https://doi.org/10.1007/s11069-007-9169-3, 2008.

Huang, W., Ding, M., Li, Z., Yu, J., Ge, D., Liu, Q., and Yang, J.: Landslide susceptibility mapping and dynamic response along the Sichuan-Tibet transportation corridor using deep learning algorithms, Catena, 222, 106866, https://doi.org/10.1016/j.catena.2022.106866, 2023.

Islam, M. A., Arrafi, M. A., Peas, M. H., Hossain, T., Hasan, M. M., Murshed, S., and Tania, M. J.: Predicting urban landslides in the hilly regions of Bangladesh leveraging a hybrid machine learning model and CMIP6 climate projections, Geosystems and Geoenvironment, 4, 100354, https://doi.org/10.1016/j.geogeo.2025.100354, 2025.

Luo, J., Zhao, Z., Li, W., Huang, L., and Zhao, W.: Landslide hazard assessment of an urban agglomeration in central Guizhou Province based on an information value method and SVM, bagging, DNN algorithm, Scientific Reports, 15, 1–15, https://doi.org/10.1038/s41598-025-86258-7, 2025.

Pascale, S., Sdao, F., and Sole, A.: A model for assessing the systemic vulnerability in landslide prone areas, Natural Hazards and Earth System Science, 10, 1575–1590, https://doi.org/10.5194/nhess-10-1575-2010, 2010.

Pascale, S., Parisi, S., Mancini, A., Schiattarella, M., Conforti, M., Sole, A., Murgante, B., and Sdao, F.: Landslide susceptibility mapping using artificial neural network in the urban area of Senise and San Costantino Albanese (Basilicata, Southern Italy), Lecture Notes in Computer Science (including subseries Lecture Notes in Artificial Intelligence and Lecture Notes in Bioinformatics), 7974 LNCS, 473–488, https://doi.org/10.1007/978-3-642-39649-6\_34, 2013.